# Insights into Telomerase/hTERT Alternative Splicing Regulation Using Bioinformatics and Network Analysis in Cancer

**DOI:** 10.3390/cancers11050666

**Published:** 2019-05-14

**Authors:** Andrew T. Ludlow, Aaron L. Slusher, Mohammed E. Sayed

**Affiliations:** School of Kinesiology, University of Michigan, Ann Arbor, MI 48109, USA; alslush@umich.edu (A.L.S.); mosayed@umich.edu (M.E.S.)

**Keywords:** *hTERT*, telomerase, telomeres, alternative splicing, network analysis, hierarchical clustering analysis, differential gene expression analysis

## Abstract

The reactivation of telomerase in cancer cells remains incompletely understood. The catalytic component of telomerase, *hTERT*, is thought to be the limiting component in cancer cells for the formation of active enzymes. *hTERT* gene expression is regulated at several levels including chromatin, DNA methylation, transcription factors, and RNA processing events. Of these regulatory events, RNA processing has received little attention until recently. RNA processing and alternative splicing regulation have been explored to understand how *hTERT* is regulated in cancer cells. The *cis*- and *trans*-acting factors that regulate the alternative splicing choice of *hTERT* in the reverse transcriptase domain have been investigated. Further, it was discovered that the splicing factors that promote the production of full-length *hTERT* were also involved in cancer cell growth and survival. The goals are to review telomerase regulation via alternative splicing and the function of *hTERT* splicing variants and to point out how bioinformatics approaches are leading the way in elucidating the networks that regulate *hTERT* splicing choice and ultimately cancer growth.

## 1. Introduction

Telomeres are specialized DNA and protein structures found at the ends of linear chromosomes made up of the hexameric repeat DNA 5′-TTAGGG_n_ [1]. The main function of telomeres is to protect the ends of linear chromosomes from inappropriate recognition as broken DNA by cellular DNA damage response proteins [2]. Telomeres prevent the recognition of chromosome ends by DNA damage response proteins by being bound by a six-protein complex called shelterin. Thus, telomeres and the shelterin complex overcome the “end protection problem”. Telomeres are also involved in determining the maximal number of times a cell can divide. Due to the inability of DNA polymerase to completely replicate the lagging strand of telomere DNA, a small (30–150 nucleotides) piece of DNA is lost with each round of replication (Figure 1). This phenomenon, known as the “end replication problem”, results in telomere shortening overtime. Upon reaching a critically shortened length, telomere uncapping and DNA damage sensing of telomeres by p53 results in growth arrest [1,2,3]. Growth arrest is triggered when one or a few telomeres become short enough to be sensed as damaged DNA, resulting in replicative senescence [4]. The limited proliferative capacity, also known as the “Hayflick limit”, of cells can act as a ‘cellular aging/timing’ mechanism in humans and other large long-lived organisms. By having a counting mechanism, cells can prevent unlimited cell growth (i.e., telomeres are short and thus sensed as DNA damage). Without such a mechanism, cells could accumulate mutations associated with cancer development. Thus, telomere shortening and replicative senescence is thought to act as a potent inhibitor of progression to malignancy [1,5].

In order to achieve immortality, cancer cells need a telomere length maintenance mechanism [6]. Nearly all cancer cells up-regulate telomerase to re-elongate or maintain telomeres by de novo synthesis of telomere repeats on to chromosome ends [1,7,8]. Although most cancer cells have detectable telomerase activity, enzyme levels vary considerably between tumors and individual cells within tumors [9]. Telomere length is also heterogenous between tumor types and within tumors [10]. Telomerase is a ribonucleoprotein (RNP) with reverse transcriptase activity that consists of two main components and several accessory proteins. The core RNP is composed of the catalytic protein subunit telomerase reverse transcriptase (hTERT) and an RNA template component (human telomerase RNA component; *hTERC*, *hTR*) that when assembled and recruited can elongate or maintain telomeres [7,11]. Telomerase is active during embryonic development but is rapidly repressed in most somatic tissues [12]. Only specialized subpopulations of transit amplifying stem/ progenitor cells are capable of transient telomerase expression post-development [1,13].

Telomerase is subject to a myriad of gene expression regulatory mechanisms. Little consensus exists in the field about chromatin environment, DNA methylation, DNA looping, promoter mutations, and transcription factor binding [14,15]. Despite the vast amount of research that has focused on transcriptional and epigenetic regulation of *hTERT*, little research has focused on the regulation of the resultant RNA molecules and co/post-transcriptional gene expression regulation [16,17]. hTERT mRNA levels are highest in embryonic stem cells, induced pluripotent stem cells and transit-amplifying adult progenitors, and lower in normal cells. Contrary to the dogma in the field, recent evidence indicates that there may or may not be a slight increase in *hTERT* mRNA abundance in cancer cells [18,19,20]. Full-length (FL) *hTERT* mRNA is the limiting factor for the formation of telomerase activity. Despite the presence of active telomerase enzymes in cancer cells and stem cells, the mRNA copy number or mRNA abundance is very low compared to other genes [21,22,23]. For instance, quantification of telomerase components has shown 5000–10,000 molecules of *hTR* in cells, while *hTERT* mRNA is expressed between 1–40 molecules per cell [21,22,23]. Although the general paradigm is that the *hTERT* is limiting for active telomerase, either component can be limiting in the formation of active telomerase [23,24]. In most normal cells, *hTR* is present in excess and thus *hTERT* is limiting. Evidence for this is the observation that *hTERT* expression is sufficient for immortalization (but not transformation) of fibroblast cells [25]. Recent evidence has demonstrated that there is a subpopulation of *hTERT* protein that is not assembled into the telomerase complexes that could be capable of maintaining telomeres. Estimates indicate that there are anywhere from 100–700 hTERT protein molecules that can interact with *hTR* in a telomerase active cell at any given time [26,27]. In order to develop better telomerase inhibitors, a more thorough understanding of hTERT gene expression regulation and function is necessary to gain insights into possible therapeutic avenues. 

Due to the lack of telomerase activity in most normal cells, besides transit-amplifying stem cells and germ line cells, and the fact that the majority (~90%) of cancer cells have telomerase activity, telomerase has been a highly sought-after cancer therapeutic target. While both public and private efforts have attempted to develop inhibitors of this enzyme, the most clinically progressed drug is an anti-sense RNA (Imeltelstat, GRN163L) of the template RNA, *hTR* [10,28,29,30]. Other small molecule drugs and vaccine-like approaches to target telomerase positive cancer cells have been attempted but have failed due to dose-limiting toxicities and other off target effects on normal cells [30,31,32,33]. Further, clinical trials of Imetlestat are still underway and this drug may be best for cancers with already very short telomeres [10]. Thus, the potential therapeutic benefits of targeting telomerase have not been realized with current strategies. The major issue with direct inhibition of telomerase activity is the long lag period that it takes to treat cells with inhibitors before telomeres are critically shortened and cancer cells begin to die [1]. Recent advances in the field, however, have led to a resurgence in interest towards finding a therapeutic window and means to inhibit telomerase/target telomere biology as a cancer therapy. For instance, the observations that certain cancer cells/tumors appear to be addicted to hTERT/telomerase as indicated by rapid telomere length-independent apoptosis, suggests that there may be other strategies to target cancer cells [34]. *hTERT* promoter somatic mutations in cancer cells also provide a new approach to targeting hTERT/telomerase positive cancer cells with minimal off target effects. Additionally, a new class of drugs called telomere uncapping drugs are showing significant benefits in pre-clinical studies. Leading the way in this class is a nucleotide analogue, 6-thio-deoxyguanosine (6-thio-dG) [35]. This nucleotide is preferentially incorporated by telomerase into telomeres, which is hypothesized to generate a mutant telomere sequence. Shelterin components cannot bind to mutant (6-thio-dG containing) telomeres which contributes to rapid telomere uncapping, DNA damage signaling at the telomeres, and cell death in telomerase-expressing cancer cells [35]. Thus, a more thorough biochemical analysis of the *hTERT* regulatory mechanisms is being sought to find new and more potent telomerase/TERT/telomere biology drugs. 

One area of gene expression regulation that has mostly been ignored is alternative RNA splicing of *hTERT*. Alternative RNA splicing has recently been observed to impact at least 95% of human multi-exon genes and serves as a mechanism to control gene expression in several evolutionary conserved ways [36]. For example, alternative splicing generates proteome diversity by making several proteins from the same transcriptional unit/gene, allowing ~20,000 genes to code for more than 100,000 proteins [37]. While gene number does not scale with organism complexity, intron number and thus splicing, does scale with organism complexity [38]. Alternative splicing of a gene can lead to proteins with similar function or even opposing functions (dominant-negative isoforms). Alternative splicing can also regulate the abundance of functional gene products by splicing to isoforms that have premature stop codons (degraded by non-sense mediated mRNA decay [39]). Ultimately, alternative splicing offers a biological mechanism utilized by cells to regulate the functional outcomes of each gene. hTERT/telomerase offers a good model gene that utilizes alternative splicing as part of its regulatory repertoire, which is of particular importance in cancer biology and stem cells. 

## 2. Alternative Splicing is Dysregulated in Cancer Leading to the Re-Emergence of Splice Variants Normally Found in Development but Silenced in Normal Cells

Alternative splicing is dysregulated in cancers [40]. Alternative splicing is regulated by the combination of cellular context, *cis*-elements, and *trans*-factor/RNA binding proteins [41]. Alternative splicing is also co-transcriptionally regulated according to the kinetic coupling model [42]. As RNA polymerase II transcribes a new pre-mRNA molecule, the spliceosome is recruited to the pre-mRNA, even docking on the C-terminal domain of RNA polymerase II and dictating the inclusion and exclusion of exons [43]. Further, the rate of RNA polymerase across a gene body, along with the chromatin environment, DNA methylation patterns, and other unknown factors, can significantly impact the alternative splicing pattern of a gene [44]. The spliceosome is a megadalton molecular machine that is composed of five small nuclear ribonucleic particle (snRNP) core components (U1, U2, U4, U5 and U6) and an additional ~700 proteins [45]. The spliceosome components are recruited with RNA polymerase II to the growing pre-mRNA and assembled in a step-wise manner at the 5′ and 3′ splice sites, branch point, and polypyrimidine tract in order to complete intron lariat formation and removal, and joining of exons in the processed transcript [45] (Figure 2). Exon joining may be constitutive, meaning the exons are always included in the mRNA of a gene or alternative (only included sometimes in the mRNA of a gene), giving rise to alternative splice variants (ASVs; [45]). Other types of splicing events can occur such as intron retention, alternative 5′ or 3′ splicing sites, alternative promoters/first exons, and alternative polyadenylation/3′ exon (Figure 2). Splice site selection is a complex process but generally the proximity of local sequence elements (*cis*) such as exonic splicing enhancers/silencers (ESE/ESI) and intronic splicing enhancers/silencers (ISE/ISI) and the RNA binding proteins in the cell at any given time dictate splicing choice (Figure 2; [46]). There are at least 700 known splicing factors/RNA binding proteins that can participate in alternative splicing [47]. We are only at the beginning of understanding the roles and regulation of splicing in normal cells and the many ways cancer cells utilize dysregulated splicing to promote growth and survival. Nearly all of the hallmarks of cancer cells have dysregulated splicing products that have been identified, including hTERT and cellular immortality [48]. These mechanistic insights may pave the way for new therapeutic avenues into treating cancer, or specific aspects of cancer cells. 

## 3. Alternative Splicing of hTERT

The reverse transcriptase component of telomerase, hTERT, is subjected to regulation by alternative splicing. It is important to note that the murine TERT (mTERT) gene is not alternatively spliced in the same fashion as human TERT (hTERT) [20]. Thus, we will focus solely on the splicing of human TERT in this review. There are 22 known splice isoforms of hTERT that have been detected in a variety of cell types [49]. hTERT is a 16 exon (15 intron) gene (Figure 3 and Table 1). hTERT consists of four major protein domains (TEN domain, RNA binding domain, reverse transcriptase domain, and C-terminal domain; Figure 3). Only the full-length 16 exon isoform of TERT that codes for a protein that can be assembled into telomerase ribonucleoproteins is capable of maintaining or elongating telomeres (Table 1; [17]). hTERT is spliced into active and inactive forms simultaneously in telomerase positive cells (i.e., cancer cells, embryonic stem cells, iPS cells, male germ line precursor cells, transit-amplifying adult stem cells, and activated immune T cells). The full-length protein coding hTERT mRNA is expressed in the range of 1–90% of the steady state transcripts depending on cell line/tissue studied [19,21,22]. 

The alternative RNA splicing isoforms that are expressed in each cell type are not well described but are likely to be tissue- and cell line-specific. The most commonly studied isoforms of hTERT result from alternative splicing in the reverse transcriptase domain (RT) between exons 5 and 9 [19,50]. Alternative splicing in the RT domain consists of splicing in regions called the alpha and beta regions (Table 1; Figure 3). The hTERT alpha region is a cryptic splice site within exon 6 that results in deletion of the 5′ 36 nucleotides resulting in the minus alpha variant [51]. This alternative variant is in the canonical hTERT reading frame and codes for a dominant-negative protein that can interact with hTR, and when overexpressed, results in telomere shortening in telomerase positive cells [51]. However, this variant is not very abundant, accounting for less than 5% of the steady state transcripts in cancer cells [22]. The beta region consists of exons 7 and 8 of hTERT and these exons are skipped in the minus beta variant of hTERT. The skipping of exons 7 and 8 of hTERT puts a premature stop codon in exon 10 in frame and thus results in the majority of the steady state mRNA of this transcript being targeted for non-sense-mediated decay [22,51]. However, recent evidence in certain cancer cells indicates that not all of this transcript is degraded and some may interact with polyribosomes and be translated into truncated hTERT proteins [50]. The suspected function of minus beta truncated hTERT proteins is similar to that of minus alpha in that it would contain exon 2 and the RNA binding domain, and thus could interact with hTR and compete with full-length telomerase for telomere binding [26]. Other evidence indicates that minus beta may be interacting with DNA damage and repair complexes and be protecting cells that express this variant from certain types of genotoxic stressors [50,52]. However, these results are controversial since an antibody to minus beta hTERT does not exist. The abundance of minus beta varies from cell type to cell type but can be anywhere from 10% to 90% of the steady state transcript levels [19,50]. The combination of minus alpha and minus beta splicing also occurs in some cell types. The abundance of minus alpha minus beta can range from 1% to 15% depending on cell type [22]. The function of this variant is assumed to be null as it should be degraded by non-sense-mediated decay pathways. 

Several other variants outside of these have been described in the literature such as the minus gamma variant, Del2, INS3, INS4, and delta4–13 (Table 1) [49,53]. The gamma deletion variant results from skipping of exon 11 and is in the original reading frame of hTERT [16]. This splicing event impacts the RT domain, is highly tissue-specific, and may act as a dominant-negative protein if it is expressed at sufficient levels in cells. Recently, the Del2 (deletion of exon 2) alternative splicing variant (ASV) of TERT was quantified in several cancer cell lines [21]. Exon 2 codes for part of the RNA binding domain of hTERT. ASVs lacking this exon would be unable to interact with hTR and thus would not have canonical telomerase activity. This variant was estimated at 40 copies per cell in certain cell lines investigated but was absent in other lines, thus its expression is tissue- and cell line-specific [44]. The authors also went on to show that this ASV could indeed code for a protein of 12 kDa; however, no function or physiological studies were performed so the function of this protein is unknown [36]. Several intron retention variants exist in hTERT as well. Many of these variants contain premature stop codons, but two variants, INS3 and INS4, have been defined to function as dominant-negative inhibitors of telomerase activity [53]. INS3 contains a 159 bp insertion of intron 14 (622–781 nucleotides) at the end of exon 14, encoding for 44 amino acids, followed by a stop codon [52]. INS4 contains a 600 bp insertion of the entire intron 14, encoding for 17 amino acids, followed by a stop codon [12]. The expression of INS3 and INS4 is tissue-specific and when expressed may account for 1–15% of the total steady state levels of hTERT mRNAs. Another recent study exploring the identity of hTERT ASVs in a variety of human cell lines discovered several new variants including delta4–13 [49]. The authors demonstrated that hTERT was transcribed in all lines investigated, even telomerase negative lines, but that the transcript in the negative lines was alternatively spliced to the delta4–13 ASV which lacks the RT domain and thus cannot produce active telomerase. The delta4–13 ASV codes for a truncated hTERT protein that seemingly interacts with WNT/beta-catenin pathway and stimulates the proliferation of cells in culture. While hTERT alternative splicing variants have been documented in various tissues and cell lines to date, technological and methodological limitations make some of the above conflicting findings difficult to interpret. Moving forward and as described below, RNA sequencing technologies and new informatic techniques will pave the way for a more thorough understanding of hTERT ASVs. 

### 3.1. hTERT Alternative Splicing during Human Embryogenesis and Development Indicates that Telomerase Activity is Regulated by Alternative Splicing

hTERT is regulated by alternative splicing during human embryonic development. During tissue development and the first phases of differentiation, hTERT is transcribed and spliced to multiple forms [54,55]. The most commonly studied isoforms arising from exons 5–9 have been documented. For example, full-length (FL) hTERT and minus beta hTERT are present along with telomerase activity during kidney development. At about week 17 of development, there is a massive shift in hTERT splicing where the full-length (exon 7/8 containing) transcript is eliminated and only minus beta remains. This shift in splicing coincides with a complete loss of telomerase activity [54]. These observations can be interpreted to indicate that alternative splicing regulates telomerase activity. However, the splicing factors that regulate the turning off of telomerase activity during tissue differentiation and specification are completely unknown. Further, the expression and splicing of other hTERT ASVs is not well studied during the differentiation and development of human tissues. This area deserves further investigation as telomerase halopinsufficiency leads to stem cell diseases and risk of early cancers in patients. Thus, further characterization of hTERT regulation in stem cells may lead to early interventions and cancer prevention. 

### 3.2. A Paradigm Shift: hTERT Is Regulated by Alternative Splicing in Cancers

A long-held paradigm in the telomere/telomerase field was and still is that hTERT and telomerase is regulated by transcription. It appeared that hTERT was transcriptionally silenced following fetal development and this was the mechanism that prevented hTERT expression and thus telomerase activity, and allowed for progressive telomere shortening that is observed in the soma [16,17]. However, recent evidence from several groups indicates that this may have been a mis-interpretation of the assays used to measure hTERT steady state transcripts. The most common assays to measure hTERT transcripts are designed to detect exons 5–9 in the RT domain, however, we now know that those exons are spliced out of most transcripts of hTERT [20,49]. Thus, previous research using primers in exons 5–9 led to missing transcripts that contained other regions of hTERT mRNA and the interpretation that hTERT is transcriptionally silenced in normal somatic cells. We and others have reported that hTERT is indeed transcribed in all cells but it is spliced to forms that do not encode for reverse transcriptase activity [18,49]. Further, exon 1 of hTERT is extremely G/C rich making it difficult to detect without using PCR additives and modified polymerases at higher than normal annealing temperatures. We have quantitated that normal cells and tissues express between 50–90% of the abundance of hTERT transcripts as cancer cells and that ultimately an upregulation of transcription of hTERT in cancer cells is minimal in terms of overall transcripts [18]. The major regulatory mechanism that leads to active telomerase and full-length hTERT production is a shift in splicing. Thus, the new working model going forward should be to understand how hTERT alternative splicing is regulated in normal cells and becomes dysregulated during the progression to malignancy, leading to tumor cell immortality. 

### 3.3. RNA Sequencing and Other Technologies to Detect hTERT Splice Variants in Cancer

The abundance and splicing of hTERT makes it difficult to detect using standard techniques. Using short read sequencing and RT-PCR to quantify and identify splicing variants leads to bias and the potential for mis-interpretations of the data [56]. The coverage and sequencing depth of short read RNA sequencing experiments can significantly mislead research concerning hTERT splicing and must be interpreted and validated carefully. New and emerging sequencing technologies and informatics tools have significantly advanced the detection and quantification of full-length cDNAs [57]. For instance, Sayed et al. 2018 demonstrated using third generation single molecule sequencing of hTERT-specific cDNA libraries that HeLa cells splice hTERT into several variants [20]. This sequencing technology was combined with informatics analysis that allowed the authors to define the identity of full-length transcripts in cells [20]. The most common variants in the libraries were identified as a very short transcript-containing exons 1, 15 and 16, a transcript splicing from exon 4 to exon 16. Other variants where detected using this method such as full-length being the second most abundant transcript identified. Interestingly, minus beta as well as Del2 were detected but these variants in their full-length context were not as abundant as previously estimated by other techniques. These newer sequencing technologies and informatics have their own sets of caveats. Improvements in reagent chemistry, library generation techniques, and analysis software that allows mapping and quantification of detected transcripts of third generation sequencing will prove advantageous over other methods for splice isoform measures.

### 3.4. Regulation of hTERT Alternative Splicing by cis-Elements and trans-Factors

The general rules of splicing regulation or the splicing code are still being elucidated; several recent efforts to understand the role of *cis*-and-*trans* elements of hTERT alternative splicing regulation have been published. Two seminal studies investigated the reverse transcriptase domain alternative splicing of hTERT [17,50]. Both groups generated minigene constructs including exons 5–9 to determine what sequence elements and *trans*-factors were responsible for the formation of full-length (containing all five exons) versus the minus beta splice variants containing only exons 5, 6, and 9. In breast cancer cells, Listerman et al. focused on the formation of minus beta. Using their minigene construct they observed that the majority of the product when in the context of breast cancer cells was the full-length variant (90%) with about 10% of the observed transcripts being minus beta [50]. Next, they undertook a small-scale cDNA screen of common splicing enhancers (Serine/Arginine-rich (SR) proteins) and splicing repressors (hnRNP proteins; Figure 4). They observed that SRSF11 promoted the alternative splicing (repressed full-length splicing; Figure 4) and formation of minus beta in their minigene. They also observed that hnRNPH2 and hnRNPL promoted full-length splicing of their hTERT minigene [29]. This study also observed that not all of the minus beta transcript was degraded by non-sense mediated decay and may make a dominant-negative protein of telomerase. Further the authors demonstrated that minus beta may protect breast cancer cells from chemotherapeutic insults [29]. Thus, future research is needed to more carefully explore the role of minus beta in cancer cells. 

In a study by Wong et al., an hTERT minigene was generated and an interesting observation was made that the initial construct only formed full-length hTERT transcripts containing exons 5–9 when placed in the context of HeLa cells [58]. To determine what sequence elements may be missing in the minigene construct, the authors performed a self-blast of hTERT exons 5–10 and observed highly repetitive sequences in these exons and introns of hTERT. To determine if these repeat regions might be important in regulation, they looked at the conservation of these repeats in species that regulate TERT similar to humans (i.e., old world primates) compared to species that regulate TERT differently (i.e., rodents). The authors found several conserved repeat regions shared between old world primates and human TERT gene loci, but these elements were lacking/missing in rodents and other shorter-lived primates. Utilizing this information, the authors inserted three of the conserved elements into the hTERT minigene and observed the expected ratio of full-length to minus beta steady state expression (i.e., recapitulating the endogenous hTERT isoform expression ratio). These *cis*-elements were termed block 6 repeats (a variable number tandem repeat in intron 6), direct repeat 6 (DR6), and direct repeat 8. The direct repeats are 256 nucleotides within intron 6 and 285 nucleotides within intron 8 respectively, and consist of 85% homologous sequences [58]. Through deletion analysis, the authors determined the impact of each element on steady state hTERT isoform expression. It was observed that the 1.1 kb VNTR (38 nucleotide repeat) termed block 6 repeats was essential for exclusion/skipping of exons 7 and 8 and production of the minus beta deletion containing transcripts. Further, DR8 was important for the formation of exon 7- and 8-containing transcripts, or potential full-length transcripts. To follow up these observations, the authors went on to show that a minimal number of VNTR block 6 repeats were needed to promote minus beta splicing (skipping of exons 7 and 8) and that blocking DR8 with an anti-sense oligonucleotide could promote skipping of exons 7 and 8, indicating that DR8 is likely a docking site for *trans*-factors [58]. In a second study, Wong et al. utilized RNA secondary structure modeling to predict how the pre-mRNA could be folding following transcription [59]. They then utilized a modified mutation complementation assay to demonstrate that the VNTR block 6 repeats could potentially form RNA:RNA pairing, making the splicing of the exon 6 5′ splice site be in closer proximity to the exon 9 3′ splice site [59]. Combined, these foundational data indicate that alternative splicing of hTERT, which is a very low abundant transcript, does not follow the typical splicing rules of more abundant transcripts. These studies determined a few *trans*-factors and the pivotal sequence elements in determination of the splicing choice of exons 7 and 8 of hTERT. 

To begin to elucidate additional *trans*-factors, Ludlow et al. used a dual-reporter minigene loss of the function screen focused on 516 splicing factors [19]. The list of RNA binding proteins was derived based on both empirically determined RNA binding proteins via literature searches and searching protein data bases (Genecards, etc.) that resulted in the curation of a list of 516 putative RNA binding proteins. Following the screen, there were 110 individual genes that resulted in a two-fold change in reporter activity. Since the goal of this initial study was to understand splicing factors/RNA binding proteins involved in the promotion of full-length TERT and telomerase activity, they focused on 93 genes that resulted in a two-fold change in minus beta to full length splicing. A systematic approach utilizing bioinformatics techniques and network analysis was then utilized to focus the analysis and narrow down the list of candidate genes (detailed below in Section 4).

## 4. Using Bioinformatics to Discover hTERT Alternative Splicing Regulation in Cancers

Following high throughput screening, target identification is an important yet difficult process. Several approaches can be taken to narrow down candidates. To begin to narrow down our list of candidate genes, we utilized a panel of well characterized lung cancer cells and developed highly quantitative droplet digital PCR measures of hTERT exon inclusion/exclusion events [19]. From these measures, we were able to segregate cell lines into high hTERT full length (FL) lines and low hTERT FL lines. Using publicly available gene expression data from the same lung cancer cell lines, we used hierarchical clustering analysis based on the expression level of the 516 splicing factors. We then compared and overlapped the minigene hits to the differential expression analysis. This analysis narrowed down the list from 93 potential candidate genes to 12 genes that were differentially expressed between high and low TERT FL lines. This led us to identify one gene, NOVA1, that was related to hTERT FL splicing in non-small cell lung cancer cells that express NOVA1. We then hypothesized that NOVA1, hTERT FL, telomerase activity, and telomere length interacted to define subsets of lung cancer cells that may be more or less similar in terms of splicing factor expression. Again, we utilized hierarchical clustering analysis based on the expression of NOVA1, hTERT FL, telomerase activity, and telomere length, and clustered lung cancer cell lines into categories expressing high and low levels of these variables. We then used differential expression analysis focusing on the expression of the 516 splicing factors and found a set of splicing genes that were differentially expressed between these high and low cell lines [19]. This analysis identified a network of genes that are related to the alternative splicing of hTERT and may lead to the identification of potential lead candidate genes for targeted therapies given hTERT/telomerase specificity to cancer. These analyses were done with a combination of in-laboratory measures and publicly available data. Other studies have done similar analyses to try to understand hTERT alternative splicing in cancer. 

Investigating the genetic landscape at the hTERT locus, a group utilized largescale analysis and fine mapping to elucidate the relationships between single nucleotide polymorphisms (SNPs) and telomere length, hTERT expression, and alternative splicing [60]. The authors combined cohorts to generate a large study population that had data on 110 SNPs in hTERT and correlated these SNPs to telomere length, hTERT expression and splicing from available RNA sequencing data. Further, the SNPs were also correlated (step-wise regression analysis) to cancer risk for specific cancers. Interestingly, an SNP in intron 4 was found to impact the alternative splicing of hTERT. The minor allele of this SNP was found to impact the splicing choice of hTERT by introducing the use of a novel alternative splice donor. In a follow up study, it was observed that this SNP generated a new splice variant termed INS1B which is a variant of a known hTERT ASV called INS1 [61]. The expression of INS1B reduced telomerase activity when the authors used oligonucleotides to switch the splicing to favor INS1B. The authors concluded that this SNP results in subtle inadequacies in telomerase activity in normal cells, which over time results in an increased risk for genome instability and cancer [61]. 

Other research has used The Cancer Genome Atlas or Pan Cancer Atlas to study telomere and telomerase biology including hTERT alternative splicing. Barthel et al. utilized these public resources to analyze a variety of regulatory features that lead to the expression of hTERT in cancer cells [62]. Concerning the alternative splicing of hTERT, the authors reported that the full length transcript was the most abundant in the samples with detectable levels of hTERT. This is in contrast to the common thought paradigm that the minus beta transcript is most abundant in cancer cells. However, more recent data and improved reagents and techniques are providing more evidence that FL may indeed be more abundant compared to commonly measured alternatively spliced transcripts. Several technical limitations should be mentioned briefly. hTERT is an extremely low abundant transcript making it difficult to detect and quantify accurately. RNA sequencing technologies are limited in the sensitivity for low abundance targets and thus caution must be taken when interpreting hTERT expression estimates from large consortia RNA sequencing data. Furthermore, predicting the functional outcome of full-length hTERT must also be interpreted cautiously. hTERT protein can be assembled into active telomerase molecules but it also has telomere-independent roles in cells [26,63]. That being said, this group attempted to derive a gene expression signature to predict telomerase activity levels [62]. This expression signature generated a telomerase activity score and was correlated to expression levels of hTERT and hTERC (telomerase RNA component) in the Pan-cancer analysis [62]. Overall, this paper utilized a wide variety of bioinformatic tools and public data to make inferences about the activation of telomerase in cancer and how hTERT and hTERT splicing may be related to cellular immortality of tumors. Very recently, a group published an additional *trans*-factor that may bind hTERT pre-mRNA to regulate the splicing choice of exons 7/8 of hTERT. Wang et al. reported that an antisense oligonucleotide aimed at the intronic cluster of SRSF2 binding sites in intron 6 of hTERT results in reduced FL hTERT splicing and increased alternative splicing [64]. These data combined indicate that alternative splicing is an emerging important regulatory paradigm for hTERT and telomerase and that it may indeed be targetable for cancer therapeutics. 

## 5. Utilizing Predictive Models of RNA Folding and RNA *trans*-Factor Binding

Alternative splicing is regulated by the combination of *cis*- and *trans*-acting factors along with the combination and competition of *trans*-RNA binding proteins available in a given cell or tissue at a given time (i.e., context). Many computational biology groups have attempted to model and predict both RNA folding in vivo and RNA *trans*-factor binding (recently reviewed in References [65,66]). There are many programs that have resulted from such efforts. Groups have utilized these programs to predict the hTERT RNA secondary structure to help explain alternative RNA splicing. Wong et al. in 2013 and 2014 utilized RNAfold to predict the potential structure of hTERT exons and introns 5–9 [59]. They observed the potential for RNA:RNA pairing within intron 6 and between introns 6 and 8. They inferred that this model could explain why cancer cells tend to skip exons 7 and 8 and allow for the joining of exons 6 and 9. Many tools have since evolved from these initial predictive models, and as machine learning capabilities improve, better and more accurate secondary structure prediction tools will become available. 

Another important consideration in the regulation of alternative splicing is the contribution of RNA *trans*-acting factors or RNA binding proteins’ involvement in site choice. Given its importance in gene expression regulation, many tools have been developed to predict RNA–protein interactions [66]. We used a series of freely available webtools to predict where NOVA1 may be interacting with hTERT. NOVA1 is an RNA binding protein involved in neuronal development [67,68,69]. It was initially described in small cell lung cancer patients with neurological complications [67]. NOVA1 was later associated with breast and lung cancers in general. The binding motif of NOVA1 is YCAY (where Y is a C or a U in RNA) and NOVA1 has been extensively characterized in neurons [70]. We experimentally determined that NOVA1 in fact interacts with hTERT pre-mRNAs at DR8 [19]. Experimental confirmation of RNA binding–protein interaction predictions is critical as RNA:protein interactions are not completely understood and are difficult to predict. Since a number of prediction models exist, utilizing several predictive models could provide a higher level of confidence of interaction when experimental techniques are not available. Overall, the RNA biologist tool kit continues to grow and many of these tools are freely available and can be found at Galaxy, RNA Galaxy workbench 2.0. 

## 6. Conclusions

Telomerase regulation in cancer cell progression is incompletely understood. The emergence of hTERT full length mRNA and telomerase activity is a multi-step process that leads to telomere length maintenance and survival of cancer cells. The role of alternative RNA splicing in the production of full-length hTERT mRNA is not completely defined. Understanding how cells choose to splice hTERT pre-mRNAs to either functional telomerase-generating mRNAs or to alternatively spliced products will inform tumor progression models of cancer. Further, hTERT is a low abundant gene with several regulatory features that make it an interesting model gene for understanding non-canonical splicing processes. Elucidating the role of alternative RNA splicing in telomerase biology will take a combination of molecular and cellular studies coupled with bioinformatics, network analysis, the generation of new tools potentially involving machine learning, and access to large cohorts of patient samples. Overall, the knowledge gained by studying the role of hTERT alternative RNA splicing in cancer cells and during cancer progression may lead to new therapeutic targets of telomere biology and could lead to novel paradigms of gene expression regulation of low abundance genes.

## Figures and Tables

**Figure 1 cancers-11-00666-f001:**
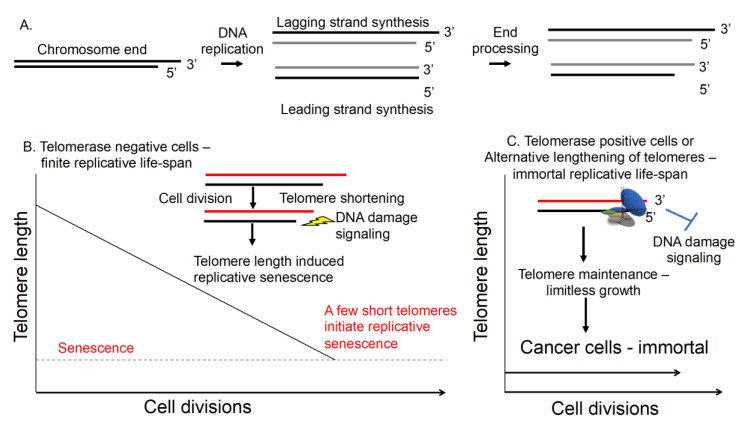
Telomere biology. (**A**) Telomeres are replicated during cell division (mitosis). A set of enzymes process the end of replicated chromosomes so that a 3′ G-rich overhang is produced. The single stranded 3′ end displaces the double-stranded structure to form a three-stranded structure (D-loop). Shelterin binds to both the single- and double-stranded portion of the telomere, protecting it from being recognized by the DNA damage machinery, solving the “end-protection” problem. (**B**) Telomerase negative cells or cells without a telomere maintenance mechanism. Due to the “end-replication” problem, a small piece of DNA at the lagging strand end of DNA is not replicated and is lost from the chromosome that is passed on to the daughter cells. Over time, this slow erosion results in the loss of telomere length. When a few telomeres have DNA damage at chromosome ends, deprotection occurs and cellular senescence is initiated. This removes cells with critically short telomeres from the replicating population of cells and acts as a potent block to tumor progression. (**C**) Cells become replicatively immortal by adopting a telomere maintenance mechanism. Telomeres are maintained by two mechanisms, telomerase RNP or a homology-directed mechanism called alternative lengthening of telomeres. Telomerase is the mechanism that approximately 90% of human cancer cells use to maintain telomeres and immortality. In male germline cells, telomeres are also maintained or elongated by the ribonucleoprotein telomerase.

**Figure 2 cancers-11-00666-f002:**
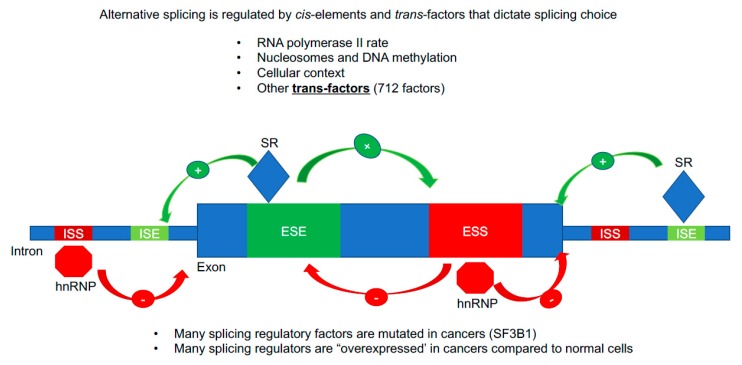
Alternative splicing regulation. A cartoon image of important sequence (*cis*) and protein (*trans*) regulatory features that result in exon inclusion or exclusion. The majority of the splicing information is contained in intronic and exonic sequences that are called intronic splicing silencers/enhancers (ISS/ISE) and exonic splicing silencers/enhancers (ESS/ESE). Specialized RNA binding proteins bind to these sequence elements and recruit in the megadalton spliceosome. Serine/Arginine-Rich (SR) proteins are typically splicing enhancers (enhanced exon inclusion) while hnRNP proteins are typically splicing silencers (repress exon inclusion, promote exon skipping/alternative RNA splicing). There are at least 700 RNA binding proteins in the human genome that can act as splicing *trans* factors, thus the repertoire of splicing regulatory features is vast.

**Figure 3 cancers-11-00666-f003:**
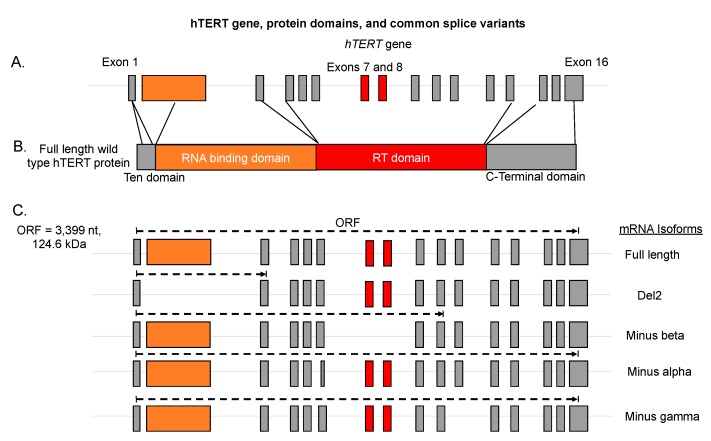
hTERT gene, protein domains and commonly studied splice variants. (**A**) Cartoon image of hTERT exons and introns. hTERT is a 16 exon/15 intron gene that generates the reverse transcriptase component of the telomerase enzyme. Exon 2 is highlighted in orange as it is the major contributor to the telomerase RNA binding domain (TRBD). Exons 7 and 8 are highlighted in red as these two exons represent one of the most commonly studied splicing events in the hTERT gene and they encode for critical residues in the reverse transcriptase domain (RT). (**B**) Protein domains of hTERT. Lines linking exons to the domains they encode are shown. Critical domains are the TEN (exon 1), RNA binding (exons 2 and 3), RT (exons 4–13), and c-terminal (exons 14–16). All four of these domains are essential for telomerase activity, processivity, recruitment, and function. (**C**) Open reading frames of abundant hTERT alternative RNA splicing isoforms.

**Figure 4 cancers-11-00666-f004:**
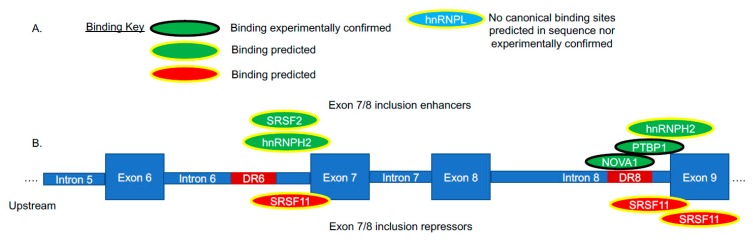
Reverse transcriptase alternative splicing regulation of hTERT. (**A**) Key of RNA binding proteins associated with hTERT. Enhancers are depicted in green. Repressors in red. Blue indicates a likely indirect impact on TERT splicing caused by the manipulation of a splicing factor. (**B**) Cartoon image of introns 5 through exon 9 of hTERT in the reverse transcriptase domain (RT). On top of the cartoon image are the hTERT exon 7/8 enhancers. On the bottom are the proteins that repress the inclusion of exons 7/8.

**Table 1 cancers-11-00666-t001:** Description of major *hTERT* splice isoforms.

Isoform	Exon Structure	Intron Retention?	Biochemical Function
**Full-length**	1–16. Original ORF.	No	Functional hTERT protein, maintains telomeres when in active telomerase holoenzyme (RNP)
**Minus beta**	1–6, 9, and 10; PTC in 10. Skipping of exons 7 and 8.	No	Mostly degraded by non-sense mediated decay, some translated into protein and may play a role in DNA damage repair/ protection from apoptosis, may bind *hTERC (hTR)*
**Minus alpha**	1–16, alternative 3′ splice acceptor site in exon 6 generates in frame shift of 36 nucleotides. Original ORF.	No	Dominant-negative, binds *hTERC (hTR)*
**INS3**	1–16 plus, PTC in intron 14.	Retention of intron 14 nucleotide 623 to end of intron 14.	Dominant-negative, binds *hTERC (hTR)*
**INS4**	1–14, and alternative exon 16 3′ splice site NT492, PTC in exon 14.	Retention of intron 14 nucleotides 1–600.	Dominant-negative, binds *hTERC (hTR)*
**DEL2**	1,3–16, PTC in exon 3.	No	Proposed mitochondrial *hTERT* variant, retains *hTERT* MLS in exon 1.
**Delta4–13**	1–3, 14–16, original ORF.	No	Proposed to stimulate proliferation. Interacts with WNT/Beta catenin.
**Minus Gamma**	Skipping of exon 11. Original ORF.	No	Tissue specific and may inhibit telomerase action at the telomeres.

PTC—premature termination codon. RNP—ribonucleoprotein. NT—nucleotide.

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
