# Peer review of "Insights into Telomerase/hTERT Alternative Splicing Regulation Using Bioinformatics and Network Analysis in Cancer"

_cancers, 2019, doi:10.3390/cancers11050666_

Round 1

Reviewer 1 Report

hTERT expression is the key step to control human telomerase activity. Exploring the mechanism of hTERT expression regulation is worthy for aging and oncogenesis. The authors described alternative splicing of hTERT and its relationship with hTERT expression in the manuscript. The alternative splicing of hTERT is an important regulator factor for hTERT transcript. This review is very helpful for fully understand hTERT expression and regulation. I recommend publishing this manuscript with the following suggestion.

1.      Telomere is progressive shortened with cell division in most human somatic cells while hTERT is repressed in these cells. But some telomerase-negative cells still can bypass the senescence and crisis then become immortal cells. So the immortal stage is not unique in the telomerase-positive cells. In figure 2, it is not appropriate to only indicate immortal in the telomerase-positive cells.

2.      The authors described the NOVA1, which regulated the hTERT alternative splicing and hTERT expression. This is a very interesting finding. But the authors talk too much about NOVA1 in unit 4 and 5. It might distract the reader’s attention on the mechanism of NOVA1 regulating hTERT expression and splicing.

Author Response

We thank the reviewers for taking the time to read and comment on our manuscript, we feel the comments have improved the content and readability. Please find our point-by-point responses to the reviewer comments outlined below in red.

Reviewer 1 comments and responses.

1.      Telomere is progressive shortened with cell division in most human somatic cells while hTERT is repressed in these cells. But some telomerase-negative cells still can bypass the senescence and crisis then become immortal cells. So the immortal stage is not unique in the telomerase-positive cells. In figure 2, it is not appropriate to only indicate immortal in the telomerase-positive cells.

-       We thank the reviewers for pointing this out. We have modified Figure 1 to include Alternative lengthening of telomeres. The figure legend is also modified to include ALT.

2.      The authors described the NOVA1, which regulated the hTERT alternative splicing and hTERT expression. This is a very interesting finding. But the authors talk too much about NOVA1 in unit 4 and 5. It might distract the reader’s attention on the mechanism of NOVA1 regulating hTERT expression and splicing.

-       We thank the reviewers for pointing this out. We have deleted the paragraph describing NOVA1’s expression profile and streamlined section 4 of the review.

Reviewer 2 Report

The present review by Ludlow et al. reports on an interesting topic and is well written/illustrated providing extensive information regarding telomerase functioning, investigation and bioinformatic analysis. Some minor issues should be addressed.

- Line 87: "that can maintain" could be changed into "capable of maintaing".

- Figure 3: legend regarding the C subpart is missing. Please check.

- Table 1: consider to also add the minus gamma variant.

- Line 314: "The" should be "They".

- Line 418: please consider to change "mutation" into "SNP".

- FL (full length): should be explained the first time is used.

- Line 446: "combine" should be "combined". Please check.

Author Response

We thank the reviewers for taking the time to read and comment on our manuscript, we feel the comments have improved the content and readability. Please find our point-by-point responses to the reviewer comments outlined below in red.

Reviewer 2 comments and our responses.

- Line 87: "that can maintain" could be changed into "capable of maintaining".

-       We have made two edits to the document.

- Figure 3: legend regarding the C subpart is missing. Please check.

- We thank the reviewer for pointing this out. We have added a description of part C of Figure 3. 

- Table 1: consider to also add the minus gamma variant.

- We thank the reviewer for this comment. We have added the minus gamma variant to both Figure 3 and table 1.

- Line 314: "The" should be "They".

            - We have corrected this typo.

- Line 418: please consider to change "mutation" into "SNP".

-       We have changed mutation to SNP.

- FL (full length): should be explained the first time is used.

-       We thank the reviewer for pointing this out and have corrected this in the document.

- Line 446: "combine" should be "combined". Please check.

-       We thank the reviewer for pointing this out and have corrected this in the document.